# THE ETHICAL AMBIGUITY OF AI DATA ENRICHMENT: MEASURING GAPS IN RESEARCH ETHICS NORMS AND PRACTICES

## ABSTRACT

The technical progression of artificial intelligence (AI) research has been built on breakthroughs in fields such as computer science, statistics, and mathematics. However, in the past decade AI researchers have increasingly looked to the social sciences, turning to human interactions to solve the challenges of model development. Paying crowdsourcing workers to generate or curate data, or 'data enrichment', has become indispensable for many areas of AI research, from natural language processing to inverse reinforcement learning. Other fields that routinely interact with crowdsourcing workers, such as Psychology, have developed common governance requirements and norms to ensure research is undertaken ethically. This study explores how, and to what extent, comparable research ethics requirements and norms have developed for AI research and data enrichment. We focus on the approach taken by two leading conferences: ICLR and NeurIPS, and journal publisher Springer. In a longitudinal study of accepted papers, and via a comparison with Psychology and CHI papers, this work finds that leading AI venues have begun to establish protocols for human data collection, but these are are inconsistently followed by authors. Whilst Psychology papers engaging with crowdsourcing workers frequently disclose ethics reviews, payment data, demographic data and other information, similar disclosures are far less common in leading AI venues despite similar guidance. The work concludes with hypotheses to explain these gaps in research ethics practices and considerations for its implications.

## 1 INTRODUCTION

When the creators of the seminal image recognition benchmark, ImageNet, pronounced that the use of Amazon's Mechanical Turk (MTurk) was a "godsend" for their research, they foreshadowed the monumental impact crowdsourcing platforms were set to have on AI research (Li, 2019). In the decade that has followed, crowdsourced workers, or 'crowdworkers' have been a central contributor to machine learning research, enabling low-cost human data collection at scale.

Ethics questions posed by research involving human participants are traditionally overseen by governance groups, such as Institutional Review Boards (IRBs) in the United States (US). Whilst medical fields and social sciences have a long history of IRB engagement, the relatively recent rise of crowdsourcing tasks in AI research means guidelines and norms have been developed in recent years to consider research ethics. The proliferation of guidelines and publication policies have risen alongside critiques of AI crowdsourced work focused on issues such as payment and worker maltreatment.

In response, this study seeks to understand how AI research involving crowdworkers engages with research ethics. It does this via an assessment of the expectations put forward by publication venues on researchers, and by analysing how these expectations translate into practices. To make this determination the policies and practices of major AI conferences, ICLR and NeurIPS, along with AI research submitted to Springer journals, are reviewed. This is compared with other benchmarks to understand whether AI research at these venues follows norms within more established disciplines. The results show that AI research at these venues involving crowdworkers lacks robust research ethics norms, with venue policies not translated into practice. Whilst ICLR, NeurIPS and Springer

provide research ethics guidance, the interpretation of these appears inconsistent, and fails to meet the same standards of disclosure as seen in other fields engaging with crowdworkers.

## 2 RELATED WORK

Oversight in research involving human subjects is no recent phenomenon, with the Nuremberg Code of 1948 formalising the idea that humans involved in research required protection (Sass, 1983; Shuster, 1997). Research ethics in the United States arose during the 1960s, prompted by various scandals in biomedical research, and followed by scandals in social science studies (Beecher, 1966; Stark, 2016; Emanuel, 2008; Heller; Milgram, 1963; Zimbardo, 1972). These cases led to regulation standardising Institutional Review Board (IRB) oversight of research involving human subjects, a requirement that exists to this day, with similar processes existing in over 80 countries globally (Grady, 2015). IRBs only oversee research involving living 'human subjects', as defined by the Code of Federal Regulations (Office for Human Research Protections, 2017).

### 2.1 CROWDSOURCING AND AI

In the twenty-first century the scope of research ethics has been extended by the rise of internet research (Taylor, 2000). The ability to recruit, engage with, and study human subjects online has led to the rise of crowdsourcing platforms, such as MTurk, becoming a key tool across a variety of academic disciplines (Howe, 2006). Launched in 2005, MTurk was an early pioneer of the crowdsourcing model (Cobanoglu et al., 2021). MTurk has remained popular due to its low cost, ease of access, and large user base (Williamson, 2016). The platform has been of particular use to the AI field, with Amazon marketing the platform as "artificial artificial intelligence" (Schwartz, 2019; Stephens, 2022).

Whilst early AI crowdwork often involved labelling tasks, such as Fei-Fei Li's seminal ImageNet work, the use of crowdworkers has diversified (Deng et al., 2009; Vaughan, 2018). Shmueli et al. offer three categories of data collection seen in NLP research papers: (1) labelling, (2) evaluation, and (3) production (Shmueli et al., 2021). For the purposes of this work these categories can be generalised across AI research.

Labelling includes the processing of existing data by a crowdworker and then the selection or composition of a label or labels for that data. Labelling can be objective, for example crowdworkers may be asked to label objects in images (e.g. dogs or cats), or subjective, with one study asking MTurk workers to label their predicted political leanings of images (Thomas & Kovashka, 2019). Evaluation involves an assessment of outputs or data according to predefined criteria, such as fluency. This could be asking humans to provide feedback on model-generated language or produce a 'mean opinion score' by assessing the outputs of various models (Clark et al., 2021; Défossez et al., 2018; Stiennon et al., 2020). Production studies ask workers to produce their own data, rather than label or evaluate existing data. For example, studies might explicitly ask crowdworkers to write questions for a question-answer dataset (Talmor et al., 2018).

These categories can be broadly encapsulated by the Partnership on AI's (PAI) definition of 'data enrichment' work, defined as data curation tasks which require human judgement and intelligence (Partnership on AI, 2021). However, this does not include research studying the behaviour of crowdworkers themselves (Vaughan, 2018). For example, a researcher might assess how individuals respond to interaction with algorithms deployed in an educational setting, or assess human perception of artificial systems (Fahid et al., 2021; Latikka et al., 2021; Koster et al., 2022). Behavioural studies are different to data enrichment tasks as they treat crowdworkers as the subject of research, rather than as a worker providing input to a model which is itself the subject of research.

### 2.2 ETHICS IMPLICATIONS OF CROWDWORK

In parallel to the rise of crowdsourcing in AI research, critics have questioned the ethics of these practices in lieu of employment law protections for workers (Aloisi, 2016). Concerns centre on issues of payment, maltreatment, power asymmetry, and demographics.

Crowdsourcing platforms are often utilised due to their low costs, and consequently many critiques of crowdwork relate to payment (Scholz, 2016). MTurk allows requesters to place tasks online

for as little as $0.01 per task, with mean payment rates estimated to be around $3 per hour (Hara et al., 2018; Pew; Toxtli et al., 2021). Considering around 75% of MTurk workers are based in the US (with 16% based in India), this is far below federal minimum wage levels (Difallah et al., 2018). This has taken focus on AI developers with the issue being reported in various media outlets (Naylor, 2021; Slater, 2021).

The rise of underpaid uncontracted work has ushered in an "era of digital sweatshops", raising concerns regarding worker wellbeing (Zittrain, 2009). An individual might be tasked with identifying harmful content, such as pornographic, violent, or offensive images, text, or video, in order to train algorithms, or might evaluate a model's moderation performance (Dang et al., 2018; Edstedt et al., 2021; Hettiachchi & Goncalves, 2019). Such subjection to harmful content has been shown to have severe psychological impacts on workers (Steiger et al., 2021; Benjelloun & Otheman, 2020).

In many cases, AI research involving crowdworkers will not include such harmful content, but that does not mean ethics concerns beyond payment are absent. Platforms have been criticised because of inadequate feedback mechanisms and intransparent instructions leading to inability to meaningfully consent to studies (Schlagwein et al., 2019; Zimmer & Kinder-Kurlanda, 2017). Some studies have been found to deceive crowdworkers, whilst researchers have the ability to reject workers outputs for unclear reasons, leading to vastly imbalanced power dynamics against workers who are often without the protection of employment contracts (Díaz et al., 2022; Irani, 2015).

The lack of consideration for workers has led to the idea that crowdsourced workers are "interchangeable" (Díaz et al., 2022). This is despite work demonstrating demographics having drastic impacts on research outcomes (Salkind, 2010; Beel et al., 2013; Welbl et al., 2021). When curating datasets with crowdworkers, lack of diversity can lead to the 'preservation' of bias in future uses of data (Celi et al., 2022; Wachter et al., 2021; Crawford & Paglen, 2021). In combination, these issues have led crowdworkers being dehumanised, and labelled "Ghost Workers" (Barbosa & Chen, 2019; Gray & Suri, 2019; Mohamed et al., 2020; Prabhu & Birhane, 2020).

## 2.3 ETHICS DISCLOSURES

How these ethics issues impact research practices has been assessed by examinations of publication practices. Santy et al. have undertaken this in the context of the NLP field, assessing how many papers engage with formal ethics review through IRBs. They find that very few papers (0.8%) cite IRB review (Santy et al., 2021). This is unsurprising considering many NLP papers will not involve any direct interaction with crowdworkers or human participants, and so would not be subject to IRB review. The authors additionally provide comparisons with other fields, such as cognitive sciences to demonstrate that NLP research lacks the same level of engagement with formal ethics processes (Santy et al., 2021).

Shmueli et al. conduct a similar review of NLP research, focusing on issues beyond payment (Shmueli et al., 2021). The authors find that whilst 10% of accepted papers at 3 major NLP conferences use crowdsourcing techniques, just 17% of these mention payment, and fewer refer to an IRB review (Shmueli et al., 2021). The paper notes that it is often unclear whether crowdworkers meet the definition of human subjects because of narrow criteria provided by US regulation (Shmueli et al., 2021). This definitional dilemma is supported by (Kaushik et al., 2022) who highlight the inconsistencies in how crowdwork is defined under research ethics regulations. These papers point to a need for further understanding of how AI researchers engage with research ethics issues.

## 3 METHODOLOGY

To assess how AI research involving crowdworkers engages with research ethics, this paper conducts (1) policy analysis of publication venues and (2) paper analysis to assess how policies and norms translate to practice at major venues.

## 3.1 POLICY ANALYSIS

The first portion of this study assesses the publication requirements of AI papers at leading venues. Google Scholar impact ratings show that conferences are the leading publication venues for AI research, and therefore the top two, the International Conference on Learning Representations (ICLR)

and the Conference on Neural Information Processing (NeurIPS) are selected for analysis (Google Scholar, 2022). Journals are also common publication venues, and can act as a comparison point for policy analysis. Springer Nature, as a leading publisher with universal disclosure policies across academic disciplines and with various journals dedicated to AI, is selected to compare policy documentation (Springer, 2022; Torres-Salinas et al., 2013).

## 3.2 PAPER ANALYSIS

The second component of this study bridges the gap between expectation and reality. In line with the policy analysis, ICLR, NeurIPS, and Springer Nature papers are assessed to understand how AI research papers adhere to the requirements prescribed by publishers. Papers were included within the study if they:

(a) Utilised data generated by humans recruited via a crowdsourcing platform[1]; and (b) Collected human-generated data directly for the purposes of the study; and (c) Contributed to artificial intelligence research

To determine whether a paper met criteria, a full-text search was performed. ICLR papers are accessible via OpenReview, a peer-review portal, whilst NeurIPS papers were accessed through the NeurIPS site. Accepted papers were then analysed via a Python script using the PyPDF package. This code identified papers which may use crowdsourcing by searching for terms such as "Mechanical Turk", "annotator" and "rater" (see Appendix A.3). Each paper identified was manually reviewed to determine whether the criteria for study was met. For Springer papers full text-search was available without additional coding via the Springer online portal. As Springer includes hundreds of journals, only papers within the Artificial Intelligence field referencing the use of MTurk, the most-used platform in the conference data analysis, were included to manage data collection. Psychology papers from Springer journals and papers from the Conference on Human Factors in Computing Systems (CHI) were also considered for one year as a benchmark of current best practices in (1) a well-established field with strong research ethics practices and (2) a conference where AI research frequently crosses into human participant research. Psychology has a long history of engagement with human participants and has similarly utilised crowdsourcing platforms in recent years, whilst CHI is arguably the leading conference sitting at the intersection of computer science and human behavioural studies (Buhrmester et al., 2018; Stewart et al., 2017).

For each paper analysed data collected included whether IRB (or equivalent) review was disclosed, whether payment terms were outlined, what type of task was undertaken, and whether any type of demographic data of workers was noted in the paper. For a full list of data points collected, see Appendix A.4.

# 4 VENUE POLICY ANALYSIS

## 4.1 ICLR

ICLR provided no requirements of authors related to research ethics until 2021, when a code of ethics was introduced. This code outlines principles such as "avoid harm", and "respect privacy" which have direct implications to research involving crowdsourcing.

The conference explicitly notes the need for the disclosure of ethics review when considering the principle of upholding scientific excellence. The code states: "Where human subjects are involved in the research process (e.g., in direct experiments, or as annotators), the need for ethical approvals from an appropriate ethical review board should be assessed and reported" (ICLR, 2022). This is the only reference to human subjects within the code, but suggests that papers should report research ethics reviews, or disclose when research ethics reviews were deemed exempt by a review body. The code does not require papers to report on other issues such as consent, payment, or demographics. Reviewers are asked to raise potential violations of the ICLR Code of Ethics, and authors are encouraged to discuss ethics questions.

---

[1]Where uncertainty exists over whether the study was conducted using a crowdsourcing platform, papers are included within analysis for completeness

## 4.2 NEURIPS

NeurIPS, a year before the ICLR Code of Ethics, piloted an ethics review process (Ashurst et al., 2022). This process focused on an assessment of the "broader impacts" of research, asking researchers to include a statement on how the research might lead to beneficial or harmful outcomes to society. The guidance for these statements was limited, and did not explicitly require disclosures of IRBs, payment rates, or consent protocols.

However, in 2021 NeurIPS replaced the requirement of an ethics statement with a checklist (Beygelzimer et al., 2021). When announcing the checklist, the program chairs stated that they aimed to "encourage best practices for responsible machine learning research, taking into consideration reproducibility, transparency, research ethics, and societal impact" (Beygelzimer et al., 2021). The checklist includes specific requests for disclosing information about crowdsourced workers or human subjects, asking the following:

"(a) Did you include the full text of instructions given to participants and screenshots, if applicable? (b) Did you describe any potential participant risks, with links to Institutional Review Board (IRB) approvals, if applicable? (c) Did you include the estimated hourly wage paid to participants and the total amount spent on participant compensation?" (NeurIPS, 2021)

This checklist shows an actionable approach to engaging with research ethics issues such as payment and ethics review, showing a clear interest in the impacts of research on crowdworkers.

## 4.3 SPRINGER

Whilst codes of ethics and requirements pertaining to research ethics are a relatively new phenomenon for AI conferences, this is not the case for journals released by established publishing houses. Publishers can have editorial policies that apply to all journals submitted, and this is the case for Springer, which publishes a number of AI journals.

Springer has a dedicated editorial policy to studies involving human subjects, stating that papers should: "include a statement that confirms that the study was approved (or granted exemption) by the appropriate institutional and/or national research ethics committee (including the name of the ethics committee) and certify that the study was performed in accordance with the ethical standards as laid down in the 1964 Declaration of Helsinki and its later amendments or comparable ethical standards" (Springer, 2022).

The policy goes on to state that any exemptions should be detailed within manuscripts, including the reasons for exemption. The publisher does not provide guidance on disclosure standards or research requirements beyond research ethics approvals, but this requirement applies to any journal submitted to Springer, meaning all papers which have engaged with an IRB should disclose this.

These policies demonstrate some deviation between the requirements of publishers across AI venues, whilst a broader analysis of publication policies across venues including ICML, AAAI, and CHI can be found in Appendix A.8, demonstrating that these policies are in-line with practices of other venues.

## 5 PAPER ANALYSIS

Whilst policy can inform how AI researchers are expected to engage with research ethics considerations, paper analysis enables an assessment of this engagement in practice.

### 5.1 CONFERENCE ANALYSIS

Table 1 show the number of papers which meet this study's inclusion criteria for crowdsourcing from NeurIPS and ICLR. For both conferences the proportion of papers meeting this criteria is low, at between 2-3% for NeurIPS and 2-6% for ICLR each year. This is a small but significant number of papers, particularly considering many more papers use previously collected datasets which may have derived from crowdsourcing, but were considered out of scope for this study.

Analysis also found that papers overwhelmingly utilised the MTurk platform compared with others. Over half of all papers using crowdsourcing at NeurIPS, and 65% at ICLR, state that MTurk was used for data collection, largely remaining consistent over the four years of data collection. Over 40% of papers at NeurIPS, and a third at ICLR, did not disclose a data collection platform, whilst the only other platform to be mentioned in more than one paper was Prolific, with four references. Full platform details can be found in the Appendix A.6.

Table 1: Proportion of accepted papers meeting crowdsourcing criteria between 2018-2021

| Year | NeurIPS Papers | NeurIPS Crowdsourcing | ICLR Papers | ICLR Crowdsourcing |
|------|----------------|-----------------------|-------------|---------------------|
| 2022 | N/A - no data | N/A | 1,068 | 24 (2%) |
| 2021 | 2,331 | 48 (2%) | 874 | 40 (5%) |
| 2020 | 1,896 | 34 (2%) | 696 | 23 (3%) |
| 2019 | 1,426 | 37 (3%) | 500 | 30 (6%) |
| 2018 | 1,009 | 24 (2%) | 334 | 11 (3%) |
| Total | 6,662 | 143 (2%) | 3,472 | 128 (4%) |

## 5.2 CONFERENCES: ETHICS DISCLOSURES

Table 2 shows analysis of research ethics disclosures for NeurIPS. In 2018 and 2019 few papers disclosed research ethics considerations. However, in 2020 18% of papers discussed IRB review, and 12% disclosed payments. This increase may have resulted from the introduction of broader impact statements, explicitly asking authors to consider the societal implications of their research (Ashurst et al., 2022). In 2021 this system changed, and a checklist was introduced explicitly requesting authors to disclose payment terms and disclose IRB reviews (Beygelzimer et al., 2021). Whilst this had a huge impact on payment disclosures, this did not significantly increase other disclosures, indicating some, but limited, success of the checklist.

Table 2: NeurIPS: Crowdsourcing papers' research ethics disclosures between 2018-2021

| Year | Crowdsourcing | IRB | Payment | Consent | Demographics |
|------|---------------|-----|---------|---------|--------------|
| 2021 | 48 | 9 (19%) | 26 (54%) | 8 (17%) | 5 (10%) |
| 2020 | 34 | 6 (18%) | 4 (12%) | 3 (9%) | 4 (12%) |
| 2019 | 37 | 0 (0%) | 4 (11%) | 1 (3%) | 4 (11%) |
| 2018 | 24 | 0 (0%) | 1 (4%) | 0 (0%) | 0 (0%) |
| Total | 143 | 15 (10%) | 35 (24%) | 12 (8%) | 13 (9%) |

In contrast, ICLR's Code of Ethics, whilst also requesting IRB disclosures, makes fewer additional disclosure requirements for papers, and this can be seen from the results in Table 3. Across 2018 and 2019 none of the 41 papers which involved crowdsourcing data collection referenced any of the categories analysed. In 2020 this changed, with some IRB and payment disclosures. Following the introduction of the Code of Ethics disclosures have become slightly more common, but remain very infrequent.

## 5.3 JOURNAL COMPARISON

The data from NeurIPS and ICLR can be compared to papers and articles submitted to Springer journals to understand whether this is a unique issue to conferences. Table 4 outlines disclosures within AI papers which utilise MTurk for data collection between 2018 and 2021. Over the four years there has been a steady increase in research ethics disclosures suggesting a trend towards AI researchers taking a greater interest in ethics considerations when using crowdworkers.

Table 3: ICLR: Crowdsourcing papers' research ethics disclosures between 2018-2021

| Year | Crowdsourcing | IRB | Payment | Consent | Demographics |
|------|---------------|-----|---------|---------|--------------|
| 2022 | 24 | 3 (13%) | 5 (21%) | 2 (8%) | 1 (4%) |
| 2021 | 40 | 1 (3%) | 5 (13%) | 2 (5%) | 3 (8%) |
| 2020 | 23 | 2 (9%) | 3 (13%) | 0 (0%) | 0 (0%) |
| 2019 | 30 | 0 (0%) | 0 (0%) | 0 (0%) | 0 (0%) |
| 2018 | 11 | 0 (0%) | 0 (0%) | 0 (0%) | 0 (0%) |
| Total | 128 | 6 (5%) | 13 (10%) | 4 (3%) | 4 (3%) |

Table 4: Springer Journals: MTurk papers' research ethics disclosures between 2018-2021

| Year | Crowdsourcing | IRB | Payment | Consent | Demographics |
|------|---------------|-----|---------|---------|--------------|
| 2021 | 76 | 16 (21%) | 22 (29%) | 14 (18%) | 26 (34%) |
| 2020 | 47 | 6 (13%) | 19 (40%) | 6 (13%) | 16 (34%) |
| 2019 | 66 | 4 (6%) | 8 (12%) | 5 (8%) | 16 (24%) |
| 2018 | 86 | 4 (5%) | 12 (14%) | 4 (5%) | 11 (13%) |
| Total | 275 | 31 (11%) | 62 (22%) | 30 (11%) | 69 (25%) |

## 5.4 PSYCHOLOGY AND CHI COMPARISON

We can compare this data to a benchmark collected from an academic field with a history of research ethics considerations, Psychology, and a Computer Science venue with a history of human data collection, CHI. Table 5 outlines the results of data collection for Psychology papers within Springer journals and CHI papers which reference the use of MTurk. The results show that Psychology papers disclose research ethics considerations most frequently, whilst CHI papers are more likely to include these compared with other AI venues, particularly when considering payment and demographic information. This could indicate either a substantive difference in the nature of the data collection between venues, or a cultural divide.

Table 5: Benchmark: Springer Psychology and CHI MTurk papers' research ethics disclosures

| Venue | Year | Papers | IRB | Payment | Consent | Demographics |
|-------|------|--------|-----|---------|---------|--------------|
| ICLR | 2018-22 | 128 | 6 (5%) | 13 (10%) | 4 (3%) | 4 (3%) |
| NeurIPS | 2018-21 | 143 | 15 (10%) | 35 (24%) | 12 (8%) | 13 (9%) |
| Springer AI | 2018-21 | 275 | 30 (11%) | 61 (22%) | 29 (11%) | 69 (25%) |
| CHI | 2022 | 66 | 28 (42%) | 49 (74%) | 27 (41%) | 41 (62%) |
| Springer Psych | 2021 | 268 | 193 (72%) | 173 (65%) | 194 (72%) | 240 (90%) |

## 5.5 TASK TYPE ANALYSIS

To explore this gap, the type of task undertaken within the research can be examined (see Table 6) to understand if disclosures differ depending on the type of engagement with crowdworkers (with task types defined in Section 2.1)[2]. 95% of Psychology papers analysed are classified as involving a 'behaviour' task, while around one in ten AI papers at AI conferences and a third of AI journal papers involve behaviour tasks. At AI conferences, evaluation tasks are most prominent, comprising 70% of all papers, whilst in journals this figure is only 39%. This may indicate that behaviour tasks are more likely to engage with research ethics issues.

---

[2]Note categories are not mutually exclusive, with some studies employing crowdworkers for multiple task types.

Table 6: Task type comparison across venues between 2018-2021

| Venue | Behaviour | Evaluation | Labelling | Production |
|---|---|---|---|---|
| AI: NeurIPS | 15% (22/143) | 70% (100/143) | 6% (8/143) | 11% (16/143) |
| AI: ICLR | 9% (9/104) | 70% (73/104) | 9% (9/104) | 16% (17/104) |
| AI: Springer | 35% (95/275) | 39% (108/275) | 17% (48/275) | 9% (24/275) |
| CHI (2022 only) | 74% (49/66) | 20% (13/66) | 12% (8/66) | 3% (2/66) |
| Psychology (2021 only) | 95% (254/268) | 2% (6/268) | 2% (5/268) | 1% (3/268) |

Table 7 explores this hypothesis, demonstrating that a gap appears to remain between the AI papers analysed and the Psychology field. However, NeurIPS disclosures are similar to those seen at CHI, except when considering payment and demographic data. The gap is most stark when considering ICLR behavioural papers, which rarely report on research ethics issues.

Table 7: Disclosures of papers utilising behavioural tasks across venues between 2018-2021

| Venue | Behaviour | IRB | Payment | Consent | Demographics |
|---|---|---|---|---|---|
| AI: NeurIPS | 15% (22/143) | 45% (10/22) | 50% (11/22) | 45% (10/22) | 45% (10/22) |
| AI: ICLR | 9% (9/104) | 0% (0/9) | 11% (1/9) | 22% (2/9) | 22% (2/9) |
| AI: Springer | 35% (95/275) | 24% (23/95) | 48% (46/95) | 23% (22/95) | 64% (61/95) |
| CHI (2022 only) | 74% (49/66) | 47% (23/49) | 80% (39/49) | 45% (22/49) | 73% (36/49) |
| Psychology (2021 only) | 95% (254/268) | 73% (186/254) | 67% (170/254) | 74% (187/254) | 94% (239/254) |

## 5.6 INSTITUTION TYPE ANALYSIS

Another explanation for this gap might be the types of institutions submitting papers, where private companies, who may be less familiar with research ethics process, could be less likely to engage with research ethics considerations. This hypothesis is explored in Appendix A.7, which demonstrates that this cannot be easily concluded, with disclosures across institutions in the papers analysed falling short of those seen at Psychology and CHI.

# 6 KEY FINDINGS

## 6.1 LEADING AI RESEARCH VENUES DO NOT ALIGN WITH TRADITIONAL RESEARCH ETHICS DISCLOSURE STANDARDS

Research ethics disclosures appear to be less common at leading AI research venues compared with Psychology and CHI. One might argue that this is due to the nature of the tasks being different, with Psychology research concerning the behaviour of participants. However, the gap persists in AI research involving behavioural tasks, whilst ethics issues are not exclusive to behavioural studies. This difference may result from how the definition of a 'human subject' is interpreted, with AI researchers unclear on when studies require engagement with research ethics review processes (Shmueli et al., 2021; Kaushik et al., 2022).

This gap may also exist because the AI field lacks the same history of engagement with human subjects, with recent crowdsourcing possibilities provoking greater interest in direct human engagement. Alternatively, this may result from a lack of research ethics education in Computer Science departments, meaning there is less focus on these concerns.

## 6.2 Journals and conferences have power to influence engagement with research ethics

The research ethics discrepancies noted are not equal across publication venues, with Springer and NeurIPS papers more frequently citing IRB reviews compared with ICLR papers. For Springer, this may be due to the varied nature of journals, many of which are related to AI's impact on society (e.g. "AI Society" and "Artificial Intelligence in Education"). For NeurIPS, a drastic increase in reporting of IRB engagement and payment terms may be explained by changes in conference policy, with the 2021 introduction of the checklist. The lack of equivalent impact of the ICLR Code of Ethics may be due to the code providing less stringent stipulations.

## 6.3 Leading AI research breaks with scientific tradition by scarcely considering demographic impacts

One type of disclosure which stands out across AI papers relates to demographic data. Demographic data is frequently reported in social science studies (including the assessed Psychology papers), from a scientific integrity and reproducibility perspective, and because of research ethics considerations (Connelly, 2013; "Nature", "2022"; Robinson et al., 2017). However, these disclosures were not seen in many of the AI papers analysed. Whilst demographics might not impact some studies (e.g. those involving objective labelling tasks), many AI research tasks involve some subjectivity, and the distinction between objective and subjective labelling is not always clean or self-evident. Demographics are frequently demonstrated to be an important influence on datasets in AI, and can have multiplicative effects as datasets are reused, locking in biases which can cause representational harms to those not adequately considered within a dataset (Paullada et al., 2020). This can be hard to identify and mitigate when datasets are re-used without demographic data, and means addressing these issues up front is critical.

The relative lack of disclosure may exist because demographic data is not collected for legal reasons or to avoid data which might be considered identifiable. However, this type of data is available for collection via crowdsourcing platforms, so this is a design choice from researchers, rather than an imposition. Studies may also argue that they do not require demographic diversity; for example if engaging in objective labelling tasks. However, labelling tasks account for 12% of papers analysed, and in many cases data enrichment tasks involve some subjectivity. Subjectivity is not the only reason to include demographic data; different demographic groups may be impacted differently by tasks, or data collected could be re-used by other actors in domains where demographic differences are impactful. Demographic reporting is an important aspect of best practices in research, and there is little excuse for the AI field to depart from this norm.

## 7 Conclusion and Future Work

This paper shows how AI researchers at leading venues engage with research ethics questions when employing crowdsourced workers, and illustrates the norms developing in the field. The work shows that research ethics disclosures are infrequent at leading research venues, whilst publication policies are emerging but inconsistently followed.

The gap demonstrated in this work between policies and practices of AI venues has been conducted with a relatively small pool of papers (owing to the content of published papers, not methodological limitations) and venues. Future work could extend this study to other venues, and should prompt further exploration of how AI data enrichment research should engage with research ethics norms which exist in other fields. While our findings are limited to the reviewed venues, they are nonetheless significant given the leading position of ICLR and NeurIPS and the role these venues play in setting research culture and norms across machine learning and AI research. This may result from ambiguity set by the regulatory requirements, which were designed for different fields and different types of work, or from a lack of experience in the AI field with this type of work. Future work could also further explore the motivations for the lack of demographic reporting, with this gap appearing consistent across the AI papers analysed, in stark contrast to research norms. With these directions in mind, this work hopes to encourage the field to move towards agreed ethics norms, fit for AI research and crowdwork, and consistently applied across publications.

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

# A    APPENDIX

## A.1    CODE

The code used in order to access and search NeurIPS and ICLR papers for this project is available on Github: [link removed for anonymity].

## A.2    RESEARCH ETHICS STATEMENT

This project was reviewed and approved by the institutions in accordance with the procedures laid down by the institution for ethical approval of all research involving human participants, reference number [redacted for anonymity].

## A.3    SEARCH TERMS

The below terms were used within a free-text search to identify papers which involved crowdsourced workers at the NeurIPS and ICLR conferences:

'mechanical turk', 'mturk', 'prolific', 'crowd', 'rater', 'annotator', 'participant', 'amt', 'labeller', 'labeler', 'figure eight'.

Figure Eight and Prolific were included to identify whether platforms other than MTurk were common in AI research, and saw different practices. Only 4 papers citing Prolific were identified, and zero citing Figure Eight, limiting this exploration.

The below terms were used within a free-text search on the Springer platform to identify Springer AI and Psychology papers which involved the use of Mechanical Turk workers:

'Mechanical turk', 'mturk'

## A.4   DATA COLLECTION CRITERIA

The below categories of data were collected for each paper examined in this study, with sub-bullets outlining the justification for data collection.

1. Is IRB or equivalent process mentioned (including disclosure of exemptions)? (Yes/No)
   - Disclosure of IRB review is a norm for human subjects research, per the Declaration of Helsinki, with IRBs ensuring the welfare of subjects in research.

2. Are payment terms for workers disclosed? (Yes/No)
   Payment is a key issue for crowdworkers within and beyond research, raising ethical and legal concerns (Felstiner, 2011).

3. Was worker consent discussed in the paper? (Yes/No)
   - Participant informed consent is a key facet of research ethics, as per the Declaration of Helsinki and Belmont Report (National Commission for the Protection of Human Subjects of Biomedical and Behavioral Research, 1978; World Medical Association, 2009).

4. What type of data collection did this work involve? (Behaviour/Evaluation/Labelling/Production)
   - Provides data on the types of tasks which AI research engages in to determine whether ethics disclosures differ between task types. Task definitions align with those described in Section 2 (Shmueli et al., 2021; Vaughan, 2018). These categories are not mutually exclusive, with some studies engaged in multiple types of tasks.

5. Were worker demographics disclosed in the paper? (Yes/No)
   - The demographics of workers can impact the outcomes of research, with research ethics reviews considering issues of representation in participant samples. This may be of particular importance in AI research, with various examples of biased data leading to unequal outcomes in AI systems (Mehrabi et al., 2022; Paullada et al., 2021).

6. What location was the institution of the lead author based in?
   - Provides insight on whether disclosures differ across geography, following the methodology outlined in Santy et al. (Santy et al., 2021).

7. In what type of institution(s) were the authors of the paper based? (University/Industry/State/Joint)
   - Provide insight on whether disclosures differ between private, academic, and state institutions, following methodology outlined in Santy et al. (Santy et al., 2021). State institutions include state-run research labs (e.g. military research bodies).

8. If disclosed, which crowdsourcing platform was used to collect data?
   - Identifies which platforms are most prominently used, and may identify variance in practices between platforms.

## A.5   GEOGRAPHIC DISTRIBUTION OF PAPERS ACROSS VENUES

The table below demonstrates the total number of AI papers assessed in this work across geographies and venues. As shown, the first authors from over half of the papers identified as using crowdsourcing across the venues derived from the US, with 13 percent from the European Union, and 9 percent from China.

This provides context to the results that follow and shows a heavy US-bias to these venues, and the study as a whole.

## A.6   PLATFORM USE

Tables below show the breakdown of platforms used across ICLR and NeurIPS papers meeting study criteria.

The table demonstrates the overwhelming reliance on MTurk for crowdsourced data, and frequency of papers choosing not to disclose platforms used.

Table 8: Geographic distribution of AI crowdsourcing papers across publication venue between 2018 and 2021

| Geography | NeurIPS | ICLR | Springer AI | Total |
|---|---|---|---|---|
| United States | 86 (60%) | 69 (66%) | 131 (48%) | 286 (55%) |
| European Union | 11 (8%) | 4 (4%) | 52 (19%) | 67 (13%) |
| China | 17 (12%) | 11 (11%) | 21 (8%) | 49 (9%) |
| United Kingdom | 6 (4%) | 3 (3%) | 10 (4%) | 19 (4%) |
| Canada | 2 (1%) | 5 (5%) | 11 (4%) | 18 (3%) |
| South Korea | 6 (4%) | 6 (6%) | 4 (1%) | 16 (3%) |
| Switzerland | 4 (3%) | 0 (0%) | 9 (3%) | 13 (2%) |
| Rest of World | 11 (8%) | 6 (6%) | 37 (13%) | 54 (10%) |
| Total | 143 (100%) | 104 (100%) | 275 (100%) | 522 (100%) |

Table 9: NeurIPS: Crowdsourcing papers' platform use between 2018-2021

| Year | MTurk | Other | Unknown |
|---|---|---|---|
| 2021 | 23 (48%) | 4 (8%) | 21 (44%) |
| 2020 | 14 (41%) | 0 (0%) | 20 (59%) |
| 2019 | 25 (68%) | 1 (3%) | 11 (30%) |
| 2018 | 15 (63%) | 0 (0%) | 9 (38%) |
| Total | 77 (54%) | 5 (3%) | 61 (43%) |

## A.7 INSTITUTION ANALYSIS

Table 11 analyses disclosures from ICLR, NeurIPS and Springer AI papers across different institution types. "Joint" indicates that co-authors on a paper represent multiple types of institution.

## A.8 VENUE POLICY ANALYSIS

Table 12 table outlines the disclosure requirement of venues assessed in this paper, plus other major AI venues, ICML and AAAI. These two venues were added as the next most influential research venues per Google Metrics.

*CHI publication policy is set by the Association for Computing Machinery (ACM), with the conference adhering to this code of conduct which includes IRB requirements. **ICML publication policy advises authors to follow the NeurIPS Code of Ethics. See: https://icml.cc/Conferences/2022/PublicationEthics

Table 10: ICLR: Crowdsourcing papers' platform use between 2018-2021

| Year | MTurk | Other | Unknown |
|------|-------|-------|---------|
| 2022 | 14 (58%) | 0 (0%) | 10 (42%) |
| 2021 | 26 (65%) | 1 (3%) | 13 (33%) |
| 2020 | 15 (65%) | 0 (0%) | 8 (35%) |
| 2019 | 18 (60%) | 0 (0%) | 12 (40%) |
| 2018 | 9 (82%) | 0 (0%) | 2 (18%) |
| Total | 68 (65%) | 1 (1%) | 35 (34%) |

Table 11: Institution type comparison between 2018-2022 for ICLR, NeurIPS and Springer AI Papers

| Institution | Crowdsourcing | IRB | Payment | Consent | Demographics |
|-------------|---------------|-----|---------|---------|--------------|
| University | 307 (56%) | 42 (14%) | 78 (25%) | 33 (11%) | 62 (20%) |
| Industry | 47 (9%) | 1 (2%) | 3 (6%) | 1 (2%) | 3 (6%) |
| State | 4 (1%) | 0 (0%) | 0 (%) | 0 (0%) | 0 (0%) |
| Joint | 188 (34%) | 5 (3%) | 24 (13%) | 8 (4%) | 18 (10%) |
| Total | 546 (100%) | 48 (9%) | 105 (19%) | 42 (8%) | 83 (15%) |

Table 12: Comparison of venue policy requirements across venues considered within this paper, plus other major AI conferences

| Venue | Code of Conduct? | Author Checklist? | IRB required? | Payment Disclosure Required? |
|-------|------------------|-------------------|---------------|------------------------------|
| NeurIPS | Yes | Yes | Yes | Yes |
| ICLR | Yes | No | No | No |
| Springer | Yes | Journal specific | Yes | No |
| CHI | Yes* | No | Yes | No |
| ICML | Yes** | No | No | No |
| AAAI | Yes | No | No | No |

