# OpenReview forum: "The ethical ambiguity of AI data enrichment:  Measuring gaps in research ethics norms and practices"
_ICLR.cc/2023/Conference — Submitted to ICLR 2023_

### Official Review · Reviewer_8fXg · 2022-10-18

**Confidence:** 3
**Correctness:** 2
**Technical Novelty And Significance:** 2
**Empirical Novelty And Significance:** Not applicable
**Recommendation:** 3

**Clarity, Quality, Novelty And Reproducibility:**

The writing is clear and the analysis is likely novel, but the insights it offers are limited. There are enough details about the methods that the experiments are likely reproducible. In my opinion, the central message of the paper ("AI research does not meet the standard of psychology research for research ethics disclosures") is not well supported by the analysis.

**Strength And Weaknesses:**

The topic is certainly interesting and I agree with the authors that this is something that should be clearly discussed (more) in the machine learning community. However, I have a few concerns. First, I am not totally sure that the topic is appropriate for ICLR (cf the call for papers). While the topic is of interest to the community, this work seems to be more appropriate for CSCW or HCOMP, venues where researchers have deeply thought about this issue. Second, there are several details that are not clear from the paper, described below:
* How is "AI research" defined? There are so many venues (HCOMP/CSCW/CHI etc.) where AI research is done and authors in my experience tend to be careful about IRB reporting. Even other conferences that are noteworthy (CVPR/ACL/FAccT/ICML/AISTATS etc.) are not taken into account here. The authors seem to focus on ICLR and Neurips because of their use of OpenReview, but this is a huge limitation and narrow view of the field.
* Related to my previous concern, there are subfields within AI/ML where authors often report the demographics of study participants. This is the case for almost all of the papers with Human-AI experiments, which likely fall in the "behavioral" tasks (and a clear definition of what is "behavior" of study participants should be given), but only some of them are published at the venues considered. The simple partition defined by the authors does not recognize this.
* There is a very thin and blurry line between what does and does not constitute human subjects research. For example, see the thorough discussion of https://arxiv.org/pdf/2206.04039.pdf. This is not discussed in the paper but is certainly relevant.
* Sample size in table 1 (and in many other) is very small. Although I think the few trends are large enough that this issue does not undermine the message of the paper, it should be mentioned in the limitations of the analysis.
* For certain applications, demographics of the participants are important. For some others, they are simply not. For example, in a task where subjects are asked to draw bounding boxes around objects, knowing the demographics of the workers is not fundamental. The authors do not seem to acknowledge this. In addition, (Lum and Isaac, 2016) and (Raghavan et al, 2019) do not mention at all crowdsourcing: They deal with biases in data collection that are totally unrelated to crowdsourcing.
* In table 8, the analysis should be done conditioning on the venue and on the type of task. The observed patterns could be explained by these factors.

**Summary Of The Paper:**

The paper reviews the information reported around human subjects experiments in articles focused on artificial intelligence published at ICLR, at NeurIPS, and in Springer Nature. The authors identity papers that use crowdsourced labor (specifically papers that meet three criteria that they define), and on this set of works they analyze whether IRB is mentioned, payment terms are outlined, workers' consent and demographics are disclosed. They then analyze how reporting varies between "evaluation" and "behavioral" tasks, and compare it with trends in psychology. They also analyze how trends were affected by policy changes at ICLR and NeurIPS, e.g., the introduction of the checklist. The authors conclude that "AI research does not meet the standard of psychology research for research ethics disclosures" and advise that demographic data, when collected, should be disclosed.


**Summary Of The Review:**

In my review, I have raised several concerns and pointed to the lack of clarity around some details. The paper is not ready for publication at the current stage. The authors may address some of my concerns in the rebuttal.

---

> ### Author Response · Authors · 2022-11-14
> **Thank you (comments enclosed & see updated paper)**
>
> Thank you so much for your feedback, questions, and considerations. Accordingly, we have updated the paper (see review updates in blue), and have responded to your questions raised below:
>
> >I am not totally sure that the topic is appropriate for ICLR (cf the call for papers)
>
> In the call for papers there is a section for "ethical considerations in ML" - our work sits in this category! We think it is important work for ICLR because we are investigating papers at this venue (amongst others), which we also believe has a large role in influencing researchers' practices.
>
> > How is "AI research" defined? There are so many venues (HCOMP/CSCW/CHI etc.) where AI research is done and authors in my experience tend to be careful about IRB reporting. Even other conferences that are noteworthy (CVPR/ACL/FAccT/ICML/AISTATS etc.) are not taken into account here. The authors seem to focus on ICLR and Neurips because of their use of OpenReview, but this is a huge limitation and narrow view of the field.
>
> Thank you so much for your thoughtful response and feedback! We selected ICLR and NeurIPS as our central venues of consideration as these venues are noted as having the greatest impact in machine learning per Google Metrics, whilst Springer AI papers are determined through the Springer platforms categorisation of AI research.
>
> We purposefully did not define ‘AI research’ in the paper because there is not widespread agreement on the boundaries of the field, let alone the boundaries of the technology (see for example debate over the definition of ‘AI’ in the forthcoming EU Artificial Intelligence Act). Rather, we use ‘AI research’ as shorthand to refer to the type of work undertaken in the venues reviewed in the paper, much of which focuses on advances in machine learning specifically, while also linking it to the broader policy and ethics debates (see response to Reviewer 3) which tend to focus on AI in the first instance.
>
> Our claims should be understood as being limited to the reviewed venues, and we have added a sentence to the limitations section reflecting this. Nonetheless, we believe our findings are highly significant for AI and ML research regardless of venue, due both to (1) the leading position of NeurIPS and ICLR in ML research per Google Metrics and (2) the culture and norms setting role played by these conferences for ML research as a field.
>
> We agree that this work could extend beyond these venues, and in response to this feedback have included data from CHI as a benchmark. This data helps show that leading venues like NeurIPS and ICLR do not meet the same standards as venues where human-computer interaction is more common. We think this is important because of the influence of these venues, and hope that this demonstrates the significance of these results, with ICLR, NeurIPS, and other general AI venues failing to meet academic standards.
>
> > there are subfields within AI/ML where authors often report the demographics of study participants. This is the case for almost all of the papers with Human-AI experiments, which likely fall in the "behavioral" tasks (and a clear definition of what is "behavior" of study participants should be given), but only some of them are published at the venues considered. The simple partition defined by the authors does not recognize this.
>
> Thank you for this feedback. In our paper we define behavioural research as those which “treat crowdworkers as the subject of research, rather than as a worker providing input to a model which is itself the subject of research” (Section 2.1). This aligns with the traditional definition of human subject research provided in the US Common Rule regulation.
>
> Our argument follows that issues such as payment, demographics, and worker welfare are not specific to behavioural studies, and the same issues arise in data enrichment studies. Despite this, data enrichment studies do not seem to tackle these issues, and we hope that the illumination of this in our paper can encourage those collecting and curating data via data enrichment, and venues where data enrichment papers are published, to prioritise these issues in future.
>
> > There is a very thin and blurry line between what does and does not constitute human subjects research. For example, see the thorough discussion of https://arxiv.org/pdf/2206.04039.pdf. This is not discussed in the paper but is certainly relevant.
>
> Thank you for sharing this paper! We agree with the authors that the definition of human subjects research is one reason for inconsistent reporting, and we hope that demonstrating this issue in paper disclosures will support the authors goals of clarifying this issue. We have added a reference to the work.

---

> > ### Author Response · Authors · 2022-11-14
> > **Review response (continued)**
> >
> > [Continuation from above response, new comment added due to character restrictions]
> >
> > > Sample size in table 1 (and in many other) is very small.
> >
> > Thanks for raising this! Unfortunately our sample size is restricted by the number of papers which engage with crowdsourcing (and we aim to capture all of these papers at the venues considered). We believe that despite the relatively small number, this is significant because the scale of crowdwork can be large, and these datasets can often be re-used in future studies (see response to Reviewer 2).
> >
> > We agree that we should acknowledge this limitation, and have updated the final section of the paper to explicitly acknowledge this - thank you!
> >
> > > For certain applications, demographics of the participants are important. For some others, they are simply not. For example, in a task where subjects are asked to draw bounding boxes around objects, knowing the demographics of the workers is not fundamental. The authors do not seem to acknowledge this.
> >
> > Thank you for raising this concern - we agree that demographics of participants may not be relevant for some studies. This is certainly true of many objective labelling tasks, and in response to this feedback we have added an acknowledgement that demographic data is not relevant for all labelling tasks. However, we would argue that demographics may play a role in many other tasks, including subjective labelling (e.g. labels of ‘offensive language’) or evaluations of superior quality of outputs. Likewise, the boundaries between objective and subjective labelling tasks are not always clear cut or self-evident; capturing demographic data for all labelling tasks could thus be recommended as a best practice to benefit future re-uses of the data where such disagreement may exist. We do not explicitly recommend this course of action in the paper, but think it could be the sort of best practice adopted by the field, and hope our paper kickstarts such conversations by showing gaps in recent disclosure practices.
> >
> > Our goal is to encourage researchers to consider whether demographics is relevant - a researcher may decide that this data is not, and state that demographic data was not collected accordingly. However, with labelling tasks only 10% of ICLR papers and 6% of NeurIPS papers, we believe in many cases demographic information should be acknowledged (or considered).
> >
> > > In addition, (Lum and Isaac, 2016) and (Raghavan et al, 2019) do not mention at all crowdsourcing: They deal with biases in data collection that are totally unrelated to crowdsourcing.
> >
> > Thank you for flagging this! The intention of these citations is to demonstrate that model outcomes can be influenced by dataset bias - this was not intended to be specific to crowdsourced data. Instead of these citations we have included references to biases located in ImageNet (collected via MTurk) which have had cascading impacts and led to ImageNet being significantly amended.
> >
> > > In table 8, the analysis should be done conditioning on the venue and on the type of task. The observed patterns could be explained by these factors
> >
> > Thank you for this suggestion - we were hoping to clarify which table you were referring to, as Table 8 in the initial version of this paper related to institutions, whilst Table 7 is conditioned on venue & task (specifically behavioural task). If it’d be helpful to show further details, perhaps in the appendix, (e.g. showing institutional type with venue and task), please let us know.
> >
> >
> > Again - thank you so much for your comments and feedback. It gave us a lot to think about and we think the paper has materially improved as a result. We'd love to hear any further thoughts or suggestions you might have on the work.

---

> > > ### Comment · Reviewer_8fXg · 2022-11-18
> > > **Response to the authors**
> > >
> > > Thank you for your detailed feedback on my review. I'll briefly summarize my position with respect to the main topic of the review, being the definition of "AI research", which is the main driver of the low score I assigned to the paper. I agree on the fact that there is quite a lot of disagreement of the type of work that this research actually entails. Exactly because of this, I believe that the authors should not attempt to generalize their results to the entire field in especially because, as I pointed out, the results are unlikely to generalize. That said, I completely agree with the message that the authors are trying to communicate. But the goal of the paper is too ambitious (again, trying to talk about trends in a very broad community) and the data from the couple of venues that are considered are not enough to support the conclusions. I wish the authors the best in revising the paper.

---

### Official Review · Reviewer_aCRE · 2022-10-24

**Confidence:** 4
**Correctness:** 3
**Technical Novelty And Significance:** 2
**Empirical Novelty And Significance:** 2
**Recommendation:** 5

**Clarity, Quality, Novelty And Reproducibility:**

The paper is clearly presented and easy to follow. The paper has somewhat novelty but lacks more deep insights. The paper should be easy to reproduce.

**Strength And Weaknesses:**

Strength:
- The topic of the paper is interesting. As crowdsourcing platforms have been widely used in AI research, the researchers in this field have not paid enough attention to the ethics in the usage of crowdsourcing. This paper will inspire the thinking of this issue.
- The authors chose the Psychology papers as a baseline, which is sound and can disclose the difference between psychology research that involves ethics as a tradition and AI research.
- The venues where the AI papers were published were chosen soundly. The findings of this research are convincing and interesting.

Weaknesses:
- The main weakness of the paper is that some findings lack deep insights and thorough discussion. For example, in 2020, the disclosure of the IRB reviews started to appear at AI conferences. Why did the research community begin to pay attention to the IRB reviews? What benefits can the IRB disclosure bring to the crowd workers and the research itself? Another example is in Table 6. For behavior research, the ethical disclosure of AI research is far less than that of psychology but for the other research type, the former is higher than the latter.  For what reason, in behavior research, the ethical disclosure of AI research is rather low?
- The venues of papers can be extended. As far as we know, there are also many AI papers that utilize crowdsourcing published in AAAI and IJCAI. Also, the papers that utilize crowdsourcing in a more complicated way can be found in WWW, CHI, and HCOMP. Why did the authors not include the paper in these venues?



**Summary Of The Paper:**

This paper investigated the ethics disclosure in the AI papers that use crowdsourcing as a data collection method published in ICLR, NeurIPS, and Springer journals in the past four years. The main finding of the investigation is that compared with the Psychology paper, ethics disclosures are far less common. Also, the paper has some other interesting findings, such as the journals and conferences have the power to influence engagement with research ethics.

**Summary Of The Review:**

This paper investigated the ethics disclosure in the AI papers that use crowdsourcing as a data collection method published in ICLR, NeurIPS, and Springer journals in the past four years. The topic of the paper is interesting. As crowdsourcing platforms have been widely used in AI research, the researchers in this field have not paid enough attention to the ethics in the usage of crowdsourcing. This paper will inspire the thinking of this issue. The paper is clearly presented and easy to follow. However, the paper lacks more deep insights. Some critical issues were not deeply touched and analyzed. Thus, it is below the acceptance bar of this venue.

---

> ### Author Response · Authors · 2022-11-14
> **Thank you - responses enclosed (and see updated paper)**
>
> Thank you so much for your review, comments and questions. We have updated the paper (see changes in blue) and have responded to your questions below:
>
> > Why did the research community begin to pay attention to the IRB reviews?
>
> Thank you for this question! The data indeed shows that IRB review does not appear in ICLR or NeurIPS papers prior to 2020, though this isn’t the case for journal articles within Springer where some disclosures did exist (albeit limitedly, and this has grown).
>
> Whilst we aren’t certain why 2020 marked a change in path, we would hypothesise that this might be caused by (1) publications such as Mary Gray’s Ghost Work (2019) which began to highlight the issues of crowdsourced work lacking acknowledgement and often being mistreated and (2) a growing interest in the ethical implications of AI research, coinciding with the rise of AI principles, and discussion of ethics (e.g. in NeurIPS broader impact statements). A few influential papers documenting awareness of ethical issues in AI/ML research were published between 2018-20. So too were hundreds of AI ethics frameworks of principles, guidelines, tenets, and similar outputs from major technology companies, government organisations, NGOs, and other public-private partnerships (e.g. Partnership on AI). Both these trends effectively established ‘AI Ethics’ as a field during that time period, which coincided with the emergence of IRB requirements in ICLR and NeurIPS in 2020.
>
> > What benefits can the IRB disclosure bring to the crowd workers and the research itself?
>
> Thank you for highlighting this question, which is at the core of our study. The goal of IRB disclosure is not for the purposes of the disclosure itself, but to ensure that IRBs, where appropriate, have the opportunity to provide guidance to researchers to consider the welfare and wellbeing of humans in research studies.
>
> By requiring such disclosures, researchers are encouraged to engage with IRB processes, and therefore engage with research ethics issues such as fair payment, worker welfare, and other issues related to the wellbeing of those involved. Disclosure practices are an expected norm in other fields (as seen in Psychology) with an established history of engagement with human participants, and we would argue that the AI field should follow similar best practices insofar as it uses human subjects for research or research-related data enrichment activities. What these best practices should be, and how they should be specified to respond to the unique aspects of AI research, is a discussion we hope this paper helps kickstart.
>
> Through researchers engaging with IRBs, crowd workers should benefit from more considered engagement from researchers, hopefully leading to better payment conditions and treatment. This should also benefit the research as IRBs can advise on issues such as participant demographics, which can have substantial impacts on the generalisability and outcomes of research.
>
> > For behavior research, the ethical disclosure of AI research is far less than that of psychology but for the other research type, the former is higher than the latter. For what reason, in behavior research, the ethical disclosure of AI research is rather low?
>
> Thank you for this feedback, however we are uncertain if we are misunderstanding the question raised here! Table 6 describes the breakdown of tasks across the venues analysed, to understand how frequent different types of tasks are undertaken at the venues analysed - it does not intend to provide details about disclosures themselves.
>
> Table 7 looks specifically at behaviour research and assesses the comparison in disclosures. Other tasks types (e.g. evaluation, production, labelling) are not assessed because whilst we could expect to see different practices for research which clearly fits into traditional definitions of behavioural research, these task types all are faced with the same definitional challenge (i.e. it is unclear if such tasks are subject to IRB oversight under the common role).
>
> > The venues of papers can be extended.
>
> Thank you for this great suggestion! Including AI venues where human interactions are more common is a great idea, and in response we have added CHI data as a benchmark (similar to the Psychology benchmark).
>
> However, we have continued to focus our paper on venues not necessarily centred on human-computer interaction, because we believe these venues should still meet the same standards as others insofar as they are using human research subjects and crowdworkers to create and curate datasets, among other things. NeurIPS and ICLR are two of the most impactful venues for research in machine learning and AI,, and play a key role in shaping the culture and norms of AI research as a whole, so studying their research ethics practices in particular strikes us as particularly important for the field.
>
> Please let us know if any further clarifications are appropriate, and thank you again for your review!

---

### Official Review · Reviewer_QRKz · 2022-10-25

**Confidence:** 3
**Correctness:** 2
**Technical Novelty And Significance:** 2
**Empirical Novelty And Significance:** 2
**Recommendation:** 3

**Clarity, Quality, Novelty And Reproducibility:**

Clarity: Clearly written.
Quality: High
Novelty: Good work addressing an important aspect of ethical disclosure,although in related work there is reference to similar work on NLP papers(Shmueli et al.)
Originality: Good work again comparing with Psychology and other journals.

**Strength And Weaknesses:**

Strengths:
- Addresses an important aspect of appropriateness of disclosure and details provided.
- Clearly demonstrates a gap.

Weaknesses:
- Although there is a requirement to disclose and they found missing reporting, it is unclear if this is significant.
- As stated in the paper, while the requirements for disclosure are present now but these requirements seem have been made in 2021-22 for ICLR and NeurIPS.But the data is from 2018-2021. Clearly, voluntary disclosure standards should or could have  be higher than what had been required previously. Hence, not sure if this study is too early.

**Summary Of The Paper:**

The paper looks at papers in AI submitted at ICLR and NeurIPS for adherence to ethical declaration standards when elements of work has been crowdsourced.This has been compared to the standards met by papers in Psychology and Springer journals.
The evidence clearly demonstrates a significant gap in disclosures across all standards including ethics reviews, payment data, demographic data.


**Summary Of The Review:**

Although the work is good and important, it is unclear to me if this is ahead of time, if the requirements for disclosure for ICLR and NeurIPS were changed in 2021-22.It is hard to interpret this data for ambiguity when there was no requirement,although authors could have volunteered it.

---

> ### Author Response · Authors · 2022-11-14
> **Thank you - responses enclosed (see updated paper)**
>
> Thank you so much for your response and feedback to our paper! We have responded below to the questions and concerns raised, and have updated the paper accordingly (see latest version, with review changes highlighted in blue):
>
> > Although there is a requirement to disclose and they found missing reporting, it is unclear if this is significant
>
> When considering significance we would argue that the findings from this work are substantively significant, and whilst the absolute numbers of papers analysed is relatively small, we have provided a comprehensive analysis of relevant papers at these journals and conferences in the years studied.
>
> We also believe this is particularly important because of the compounding impacts of data collection practices. The papers we assessed reflect the point at which datasets are being created or curated through labelling or other forms of enrichment. This is not, however, the only time when these datasets will be used; datasets or models trained on them are very frequently re-used by scholars both at the venues studied and beyond. The significance of our findings is thus much larger than the raw number of papers would suggest; whilst the number of papers assessed are small, we consistently see datasets collected with human workers re-used in future studies across ML and AI research.
>
> This means that good and bad practices in the collection of data, including disclosures about ethically relevant features of the datasets creation and curation, can have knock-on impacts and become embedded into models. For example, ImageNet would have counted as just one paper in our study, but this data has been re-used hundreds of times, and the impact of any ethically questionable collection, curation, or disclosure practices in the creation of ImageNet is likewise amplified at this scale. Issues like the demographics of data labellers have been shown through prior research to influence their completion of subjective labelling tasks. As a result,  not disclosing this information can have knock-on impacts for future studies which use this data because they will be unaware of potential biases in labelling.
>
> But the significance of the study is not just a question of awareness and mitigation of potential biases in the ML pipeline. Overall, we believe that this work is significant because of the key role that human annotators, raters, and participants now play in machine learning R&D. Disclosure of how crowd  workers are treated is materially relevant from the perspective of improving research ethics best practices in the field. For example, if there is pressure to disclose payment terms, researchers might reconsider following historic average payment terms which have been as low as $3/hour (as discussed within paper)
>
> Nonetheless, we fully agree that the number of papers in the study is relatively small, and, following your feedback, have added this to our concluding section. We hope that this work encourages future data collection with a wider range of venues, and further data can be collected over time, to further explore the challenges highlighted in the paper.
>
> > As stated in the paper, while the requirements for disclosure are present now but these requirements seem have been made in 2021-22 for ICLR and NeurIPS.But the data is from 2018-2021
>
> Thank you for raising this question - requirements for NeurIPS and ICLR have indeed been updated (for both the conferences in 2021).
>
> We believe that capturing data from before the policies changed and after helps demonstrate the effectiveness of policies, and how policy changes may be implemented. To mitigate the longevity concern raised, we have added in data from ICLR 2022 which is now available, which is consistent with our existing conclusions. NeurIPS 2022 data was not available at submission in a machine readable/accessible format, so could not be included. Thank you for noting this - we hope this amendment strengthens the value of this paper!
>
> Again - thank you so much for your review and comments - please let us know if you have any further comments or suggestions on how we can improve this submission.

---

### Official Review · Reviewer_rXNM · 2022-10-27

**Confidence:** 5
**Correctness:** 3
**Technical Novelty And Significance:** 4
**Empirical Novelty And Significance:** 4
**Recommendation:** 10

**Clarity, Quality, Novelty And Reproducibility:**

The paper is very clear, easy to read, and compelling. The arguments and studies are laid out in a narrative structure that is easy to follow.

While there is no algorithm proposed, the paper is very novel and unique. Papers about our research processes, rather than the research itself, are rare but often the most important.


**Strength And Weaknesses:**

Strengths

The paper tackles a very timely topic with a thorough, well-designed study of the AI community in comparison to psychology. Choosing psychology as a baseline is well founded, as it performs much HSR and has an established practice of rigorous disclosure requirements in its publications.

The paper reveals important and clear differences between ICLR and NeurIPS, largely stemming from NeurIPS adoption of an explicit checklist of ethics considerations for authors and reviewers. While ICLR has a similar ethics policy as NeurIPS, the introduction of explicit mechanisms to state and review compliance appears to make a huge difference.

Section 6.3 makes an excellent point about the lack of consideration and disclosure of demographic data in AI. This gap has become a source of mistrust of the AI community by the general public in high-profile cases regarding bias in facial recognition systems, for example.

Weaknesses

The paper has few weaknesses, and these are minor points.

In 4.1, it would be clearer to state that ICLR does not have an explicit ethics criterion as part of its review process. The reality is that many or most authors and reviewers do not read conference policy documents, nor are they necessarily aware when these are changed. However authors and reviewers will pay much attention to requirements that are encoded into the submission and review processes, such as a checklist that must be filled out, and associated text.

There is no general discussion of ethics as a review criterion for conference papers, but this is a major recent development that should lead to greater enforcement of ethics policy. It would strengthen the paper to include a table of major AI conferences vs. ethics policies and how they are enforced, even for those that are not included in the detailed study such as CVPR, ICML, AAAI.

The finding that conference papers “overwhelmingly utilised the MTurk platform compared with others”, sec. 5.1, could be biased by the nature of the paper selection criteria, which searched for MTurk explicitly. What about papers that leveraged labeling service companies, which are increasingly abundant and often comparable in price to MT?

It would be helpful to add a summary table in sec. 5.4 with comparative numbers between the two conferences and the journal so that the reader does not have to skip back and forth between tables on different pages in order to see their differences.

A major difference between AI and psychology research is the much higher proportion requiring IRB review in the latter, as shown in Table 5. This difference could explain virtually all of the other differences, as IRB review forces researchers to consider many criteria such as potential harm to participants that may not be considered thoroughly otherwise. Data labeling tasks are usually not HSR in the US, and many researchers (correctly) do not seek IRB review for them. Considering only behavioral tasks, table 7, is an insightful way to break down the data as such tasks are more likely to require IRB review. However, I don’t think the data supports the statement that “AI behaviour studies still do not meet the same standards as Psychology experiments” for NeurIPS. IRB review was reported for 45% vs. 73% in psychology, but this could easily reflect a true difference in the nature of the behavioral science being conducted. For ICLR and AI in general the statement is more supportable.


**Summary Of The Paper:**

The paper presents a study of how ethics considerations related to crowdsourcing are disclosed in the AI community compared to psychology. The study includes ICLR, NeurIPS and Springer journals, which is a reasonable representative sample of AI publications. The primary findings are that AI lags behind psychology in ethics disclosures, but has improved greatly since 2018 particularly at NeurIPS. The paper is unique, important and very clearly written.

**Summary Of The Review:**

This paper is important and would likely generate much discussion and debate at the conference. The study could be deeper by considering more conferences, but in its current form it is sufficiently justified and complete to warrant publication.

---

> ### Author Response · Authors · 2022-11-14
> **Thank you!**
>
> Thank you so much for your review and your detailed comments and suggestions to help improve this work! We've gone through these and responded accordingly - see details below and a new attached paper with amendments highlighted in blue.
>
>
> > In 4.1, it would be clearer to state that ICLR does not have an explicit ethics criterion as part of its review process
>
> We have updated section 4.1 to clarify the expectations of ICLR reviewers, and noted that in the review guidelines there is a request for reviewers “to raise potential violations of the ICLR Code of Ethics” .
>
>
> > It would strengthen the paper to include a table of major AI conferences vs. ethics policies and how they are enforced, even for those that are not included in the detailed study such as CVPR, ICML, AAAI
>
> Thank you for this suggestion, we agree this would be of great value for the paper! Accordingly, we have added this comparison chart to Appendix and referenced this within the paper.
>
>
> > What about papers that leveraged labeling service companies, which are increasingly abundant and often comparable in price to MT?
>
> Thank you for this question - in the study design this was a bias we were acutely aware of and tried to mitigate for. When searching for ICLR and NeurIPS papers we searched for other specified platforms (i.e. Prolific and FigureEight), but found very few of these platforms referenced in papers. To go further, we also searched for general terms related to data collection (e.g. “rater”, “crowd”, “participant”, “labeller”, see full list in Appendix) to try to identify other platforms. We identified very few references to other platforms, so concluded that MTurk was by far the most popular. These figures are disclosed in the Appendix.
>
>
> > It would be helpful to add a summary table in sec. 5.4 with comparative numbers
>
> Great idea - this summary table has been added.
>
>
> > I don’t think the data supports the statement that “AI behaviour studies still do not meet the same standards as Psychology experiments” for NeurIPS.
>
> Thank you - there certainly is a larger gap between ICLR and Psychology compared with NeurIPS, so this wording has been clarified.
>
>
> Again - thank you so much for your feedback and if you have any other suggestions or comments, please do let us know!

---

### Author Response · Authors · 2022-11-18
**Revised manuscript available**

Thank you to the reviewers for their helpful feedback and suggestions - we've responded to their comments individually and uploaded an updated version of the manuscript incorporating suggestions.

These changes include:
- The addition of a new benchmark, CHI, to assess how major machine learning venues (ICLR, NeurIPS, Springer journals) compare to an conference at the intersection of human interaction and AI.
- The extension of data for ICLR to include 2022 data.
- The addition of a comparison chart of policies across major conferences, including ICLR, NeurIPS, ICML, CHI and AAAI.
- An update to the conclusion and limitations section to discuss the number of papers included in analysis.

We hope these amendments strengthen our work, and would value any other feedback and responses from reviewers.

---

### Decision · Program_Chairs · 2023-01-20

**Decision:**

Reject

**Justification For Why Not Higher Score:**

Although the study is important, its findings are similar to that of (Santy et al., 2021) and (Shmueli et al., 2021). The sample set used in the paper to represent AI venues might not be a representative set to make generalizations as the authors used data from ICLR and NeurIPS, which recently adopted Ethics Guidelines or Code of Ethics. This study might be a bit early to be able to make generalizations given in the paper.

**Justification For Why Not Lower Score:**

N/A

**Metareview: Summary, Strengths And Weaknesses:**

The paper presents a comparative analysis of ethics disclosure practices and norms applied in AI and Psychology research dissemination venues such as ICLR, NeurIPS, Springer Journals, and CHI around human subject experiments carried out by crowdsourcing workers on MTurk by analyzing disclosed information on published/submitted papers selected based on specific criteria. The paper concludes that ICLR and NeurIPS does not meet the ethics disclosure standards of Springer Journals on Psychology and CHI, which have applied ethics policy on human subject research for a long time. However, the findings show that NeurIPS is doing better than ICLR due to its earlier adoption of a policy related to ethics on human subjects in 2021.

Although the study is important, its findings are similar to that of (Santy et al., 2021) and (Shmueli et al., 2021). The sample set used in the paper to represent AI venues might not be a representative set to make generalizations as the authors used data from ICLR and NeurIPS, which recently adopted Ethics Guidelines or Code of Ethics. This study might be a bit early to be able to make generalizations given in the paper. It would be interesting to see an analysis of the rate of adoption of "Ethics Guidelines" or "Code of Ethics" in ICLR and NeurIPS and how it compares to other disciplines in the next five years.